# How Effective Is a Late-Onset Antihypertensive Treatment? Studies with Captopril as Monotherapy and in Combination with Nifedipine in Old Spontaneously Hypertensive Rats

**DOI:** 10.3390/biomedicines10081964

**Published:** 2022-08-12

**Authors:** Christina Hawlitschek, Julia Brendel, Philipp Gabriel, Katrin Schierle, Aida Salameh, Heinz-Gerd Zimmer, Beate Rassler

**Affiliations:** 1Carl-Ludwig-Institute of Physiology, University of Leipzig, 04103 Leipzig, Germany; 2Institute of Pathology, University of Leipzig, 04103 Leipzig, Germany; 3Department of Pediatric Cardiology, Heart Centre, University of Leipzig, 04289 Leipzig, Germany

**Keywords:** old SHR, antihypertensive therapy, blood pressure monitoring, LV hypertrophy, ECM markers, cardiac fibrosis

## Abstract

Background: A major problem in the treatment of human hypertension is the late diagnosis of hypertension and, hence, the delayed start of treatment. Very often, hypertension has existed for a long time and cardiac damage has already developed. Therefore, we tested whether late-onset antihypertensive treatment is effective in lowering blood pressure (BP) and in reducing or even preventing left ventricular hypertrophy and fibrosis. Methods: Twenty-one male 60-week-old spontaneously hypertensive rats (SHR) were included. Fourteen rats received oral treatment with captopril (CAP) either as monotherapy or combined with nifedipine (CAP + NIF) over 22 weeks. Seven untreated SHR served as controls. We examined the therapeutic effects on BP, heart weight and histological and biochemical markers of left ventricular remodeling and fibrosis. Results: At 82 weeks of age, BP was reduced in the CAP and CAP + NIF groups by 44 and 51 mmHg, respectively (*p* < 0.001), but not in untreated controls. Despite the late therapy start, cardiac hypertrophy and fibrosis were attenuated compared to controls. Both treatments reduced heart weight by 1.2 mg/g (25%, *p* = 0.001) and collagens I and III by 66% and 60%, respectively (*p* < 0.001), thus proving nearly equivalent cardioprotective efficacy. Conclusion: These data clearly emphasize the benefit of antihypertensive treatment in reducing BP and mitigating the development of cardiac damage even when treatment is started late in life.

## 1. Introduction

Arterial hypertension is the main risk factor for morbidity and mortality of cerebrovascular and cardiovascular diseases. The number of hypertensive adults aged 30–79 years has almost doubled in the last 30 years to more than 1.2 billion people worldwide, but only 59% of women and 49% of men with hypertension are diagnosed, and another 11–12% are diagnosed but not treated [1]. The problem of inadequate antihypertensive therapy is exacerbated by the patients’ inconsistent drug intake and inconsequent blood pressure control by the patients and their therapists. In less than 40% of diagnosed patients, hypertension has been controlled to below 140/90 mmHg [1]. Untreated or insufficiently treated arterial hypertension leads to cardiac hypertrophy and may progress to cardiac remodeling, fibrosis, dilatation, and in the final stages, to heart failure. A long history of undetected and untreated hypertension is a risk factor as high as advanced age, particularly if hypertension has persisted for some time and the first cardiac sequelae may have already occurred. The earlier in the course of pathophysiological development an efficient antihypertensive therapy is started, the more successful it can be in preventing complications [2,3].

Spontaneously hypertensive rats (SHR) are an established animal model for studying the causes and development of hypertension. This is due to their prehypertensive stage and their clinical complication pattern, which is similar to that of humans including LV hypertrophy and dilatation and often resulting in heart failure [4]. Upregulation of renin-angiotensin-aldosterone-system (RAAS)-related key genes and increased intracellular calcium concentration belong to the main factors responsible for the increased blood pressure (BP) in adult SHR [5,6]. Correspondingly, in numerous studies in SHR, profound antihypertensive and cardioprotective effects were achieved by antagonizing the RAAS and reducing intracellular calcium concentration [2,7,8,9]. The antagonization of angiotensin II (AT II) and the blockade of calcium channels have also proved to be effective antihypertensive therapies in humans and are recommended by the guidelines of the European Societies of Cardiology and Hypertension as preferred treatments of human hypertension [10]. In particular, the combination of two antihypertensive drugs is highly recommended, as better antihypertensive effects can be achieved with lower drug doses, and thus with fewer side effects and better tolerability [10,11].

One of the major problems in human hypertension and its therapy is the fact that hypertension is often diagnosed late in life when the BP has been elevated for many years. Many of those patients present an advanced degree of hypertension, and the first cardiovascular complications have already developed. The therapy has to start immediately with strict control of patients’ compliance and BP development. However, even if antihypertensive therapy is well-adjusted and controlled, a late-onset treatment may be less effective than a treatment started in the very early stages of hypertension. Studies in SHR showed that antihypertensive treatment starting in a very early phase of life (between four and 10 weeks of age) reduced blood pressure (BP) even to normotension [3,12] and prevented left ventricular hypertrophy and fibrosis [7,13,14]. In contrast, the same treatment starting at a later stage of life (24 to 30 weeks of age) had a much lower antihypertensive effect [3,15]. For treatment of human hypertension, it is important to clarify whether and to what extent a late-onset antihypertensive therapy is able to significantly reduce BP and, even more importantly, to attenuate cardiac hypertrophy and fibrosis and to delay transition into cardiac failure.

We investigated these questions in an animal model of old SHR. Rats aged 24–30 weeks can be considered to be adult. At this age, normotensive rats are still in the first half of their regular life span, which is estimated to be 2.5–3.5 years [16]. Our interest was focused on an antihypertensive therapy starting in the period of senescence. The present study was designed to investigate the efficacy of such a late-onset antihypertensive therapy on BP development and on cardiac remodeling in 60-week-old SHR. In this context, two therapeutic regimens, a monotherapy and a combination therapy, were compared. In a preceding study on seven-week-old SHR, we systematically tested the antihypertensive and cardioprotective effects of several classes of antihypertensive drugs as single-drugs and as combination treatments. Specifically, captopril (CAP) as an antagonist of the RAAS and nifedipine (NIF) as a calcium channel blocker were applied. CAP as monotherapy and a combination of CAP and NIF proved to be the most effective treatments with respect to BP lowering and to preventing cardiac hypertrophy and fibrosis [14]. These two therapeutic regimens were chosen for the present study to treat 60-week-old SHR over a period of 22 weeks. We investigated BP, heart weight (HW), biochemical markers of cardiac hypertrophy and cardiac fibrosis as well as cardiac histology.

We hypothesized that these treatments would decrease BP as well as biochemical and histological markers of cardiac hypertrophy and fibrosis in old SHR as well. The results of this study might have importance for antihypertensive treatment in humans. In particular, one of the main reasons for patients’ poor adherence to antihypertensive therapy is that they do not understand the serious consequences of chronically elevated BP. The results of studies like this may help to improve the education of patients and the strict control of antihypertensive therapy.

## 2. Materials and Methods

### 2.1. Animal Model

All experiments were performed on 21 male SHR supplied by Charles River, Sulzfeld, Germany. The animals were fed a standard pellet diet (Altromin C100, Altromin GmbH, Lage, Germany) and had free access to tap water. All animal protocols were approved by the state agency (Landesdirektion Sachsen, number and date of approval: TVV 36/08; 13 May 2009) in accordance with the Guide for the Care and Use of Laboratory Animals published by the National Institutes of Health and with the “European Convention for the Protection of Vertebrate Animals used for Experimental and other Scientific Purposes” [17].

### 2.2. Experimental Protocol

The 22-week study phase was preceded by a two-week adaptation period to accustomize the animals to drug-free tablets and to the procedure of BP measurement. The animals were 60.5 ± 0.25 weeks of age (BW 404 ± 31 g) at the beginning and 81.9 ± 0.44 weeks at the end of the 22-week study phase. They were subdivided into three groups (*n* = seven per group). The group size was calculated for a medium effect size of systolic blood pressure (SBP) reduction (with Cohen’s f = 0.25). Two groups received antihypertensive treatment with CAP (60 mg kg^−1^ d^−1^, Axxora, Lörrach, Germany) or CAP + NIF (CAP 60 + NIF 10 mg kg^−1^ d^−1^, Sigma-Aldrich Chemie, Steinheim, Germany), while the third group received drug-free tablets and served as untreated control (CTRL). The drugs were added to commercially available rodent sweets (Vitakraft-Werke, Wührmann & Sohn GmbH & Co., KG, Bremen, Germany) and formed to tablets. The tablets were given into the cages for oral uptake along with chow once daily between 9:00 a.m. and 10:00 a.m. After the two-week adaptation period, non-invasive BP measurements were carried out every two to three weeks. At the end of the experimental period, animals were sacrificed, and their hearts were removed for further analyses.

### 2.3. Non-Invasive Blood Pressure Measurement

SBP was measured non-invasively using the tail-cuff-method (TSE Blood Pressure Monitor, Series 209002, TSE Systems GmbH, Bad Homburg, Germany). Measurements were performed in awake animals about 1 to 2 h after the administration of the drugs or the drug-free tablets. The animals were placed on a heated plate (36 °C) and were allowed to move relatively freely, and were only held by the experimenter’s hand on their back or tail. We avoided the use of a conventional restraint box to minimize stress to the animals. After six to eight preliminary tests, the animals were familiar with the environment and the experimenters. The procedure of SBP measurement was well tolerated by the animals. To ensure reproducible results, for each SBP measurement a mean was calculated from two to three tests with each test containing 10 single readings.

### 2.4. Further Analyses on Heart Tissue

After extraction, HW was determined as a measure of cardiac hypertrophy. As body weight (BW) developed differently in the SHR groups, the ratio of HW/BW was calculated by relating HW to baseline BW. The apex was separated and fixated in formalin for histological examination. Pieces of the LV were frozen and stored at −80 °C for biochemical analyses.

### 2.5. Ribonuclease Protection Assay

A ribonuclease protection assay was performed as previously described [14] to determine mRNA expression of atrial natriuretic peptide (ANP), transforming growth factor-β_1_ (TGF-β_1_), TGF-β_2_ and TGF-β_3_, matrix metalloproteinase 2 (MMP-2), tissue inhibitor of metalloproteinases 2 (TIMP-2) and collagen types I (Coll I) and III (Coll III) in the LV.

In brief, total RNA was isolated according to the method of Chomczynski and Sacchi [18] using TRIZOL (Invitrogen GmbH, Karlsruhe, Germany) as described in the manufacturer’s protocol. For investigation of TGF-β, we isolated 2.5 μg of total TGF-β RNA, and for the extracellular matrix (ECM) markers ANP, MMP-2, TIMP-2, Coll I, and Coll III, 5 μg of total RNA was isolated. The isolated RNA was hybridized overnight with the template sets rTGF-β and ECM-3, respectively, and labelled with an RiboQuant^®^ In vitro Transcription Kit (Pharmingen, Hamburg, Germany) and [α-32P]-UTP as described by the manufacturer. After hybridization and digestion of the unprotected probes, the protected radioactive RNA was displayed on a denaturing polyacrylamide gel. For densitometric evaluation we used the Molecular Imager (BioRad, München, Germany). mRNA expression was semi-quantitatively determined in percent of glyceraldehyde-3-phosphate dehydrogenase (GAP-DH) mRNA.

### 2.6. Zymography

LV myocardial MMP activity was measured as described elsewhere [19]. Briefly, extracellular proteins of frozen tissue were extracted by the addition of extraction buffer and then directly used for electrophoresis. The gels were run at 200 V at 4 °C. Following electrophoresis, the gels were stained in Coomassie Blue G-250 and scanned using the EagleEye II imaging system (Stratagene, Heidelberg, Germany) for relative lytic activity after normalization to the amount of protein extract loaded onto the gel. The protein activity was related to the median of the CTRL animals (CTRL = 1.0). For more details see [20].

### 2.7. ELISA

To determine the protein content of TGF-β_1_ and TGF-β_2_ in the LV, the immunoassay was performed according to the Quantikine^®^ ELISA Mouse/Rat/Porcine/Canine TGF-β_1_ Immunoassay Protocol (R&D Systems, Inc., Minneapolis, MN, USA) and the Quantikine^®^ Human TGF-β_2_ Immunoassay Protocol (R&D Systems, Inc., Minneapolis, MN, USA). Polyclonal secondary rabbit-antibody (Dianova, Hamburg, Germany) conjugated to horseradish peroxidase specific for TGF-β_1_ was used. Absorbance was measured at a wavelength of 450 nm. The TGF-β_1_ and TGF-β_2_ concentrations were calculated by the I-smart program (Packard Instruments Company, Inc., Downers Grove, IL, USA). The protein concentrations were related to the median of the CTRL animals (Ctrl = 1.0).

### 2.8. Histological Investigation of the Heart

The apex of the heart was cut off, fixated in formalin and embedded in paraffin. Sections of 8–9 μm thickness were stained with hematoxylin—eosin and Masson’s trichrome. For the evaluation of cardiac fibrosis, sections were examined with an Axioscope microscope (Carl Zeiss, Oberkochen, Germany), digitized with AxioCam MRc 5 (Carl Zeiss, Oberkochen, Germany) and evaluated using the Photoshop CS6 image processing program. A score ranging from 0 to 3 was applied to assess the histological degree of fibrosis: 0 = no signs of fibrosis; 0.5 = marginal perivascular fibrosis and/or marginal interstitial fibrosis; 1 = perivascular fibrosis + mild interstitial fibrosis; 2 = perivascular fibrosis + moderate interstitial fibrosis; 3 = fibrosis of the entire heart [14].

### 2.9. Statistical Analysis

Statistical analyses were carried out with the software package SigmaPlot Version 14.0 (Systat Software GmbH, Erkrath, Germany) for Windows. The groups were statistically compared using Analysis of Variance (ANOVA) procedures. Firstly, a Shapiro-Wilk test of normality was performed. If data were normally distributed, we used a One Way ANOVA with a post-hoc test according to Fisher’s LSD method. This applied to SBP, HW and quantitative evaluations from histology. These data are presented as means ± SEM. If the data were not normally distributed, a Kruskal-Wallis ANOVA on ranks with a post-hoc test according to Dunn’s method was applied. These data (the results of the biochemical analyses) are given as medians [25th/75th percentile]. As age, BW and SBP at baseline might have an effect on the results, we additionally performed an analysis of covariance (ANCOVA) to assess the influence of these covariates and their interaction with treatment effects on the results. *p* values < 0.05 were considered significant.

## 3. Results

### 3.1. Systolic Blood Pressure

Baseline SBP as measured at 60 weeks of age was similar in all three groups (CTRL 231 ± 9 mmHg, CAP 228 ± 13 mmHg, CAP + NIF 245 ± 8 mmHg). During the observation period, SBP increased slightly in CTRL rats (239 ± 5 mmHg; *p* > 0.05). CAP and CAP + NIF induced a significant SBP reduction by about 20% after 22 weeks of therapy (*p* < 0.001; Figure 1). SBP measured at 82 weeks of age was not dependent on age, BW and SBP at baseline.

### 3.2. Cardiac Hypertrophy

#### 3.2.1. Heart Weight

At the end of the experimental period, the HW of 82-week-old CTRL rats was 1860 ± 93 mg. Both forms of therapy counteracted the development of cardiac hypertrophy and reduced HW significantly by 26% (CAP) and 23% (CAP + NIF). ANCOVA revealed that the treatment effects on HW did not depend on age, BW and SBP at baseline. HW/BW in untreated SHR was 4.7 ± 0.25 mg/g and was 34% higher than in the treated animals (3.5 ± 0.2 mg/g in both groups, *p* = 0.001; Figure 2a). Notably, the final BW of CTRL rats at 82 weeks of age was 378 ± 37 g and was about 15% lower than that of treated SHR (*p* = 0.08; data not shown).

#### 3.2.2. ANP mRNA Expression

ANP mRNA expression in the LV of untreated 82-week-old SHR was 264 [154/294] percent of GAP-DH mRNA expression. Both treatments resulted in a significant reduction of ANP mRNA expression to 60 [48/68] (CAP) and 43 [15/78] (CAP + NIF) percent of GAP-DH (*p* < 0.001; Figure 2b). ANP mRNA expression was not dependent on the covariates age, BW and SBP at baseline.

### 3.3. Cardiac Fibrosis

#### 3.3.1. Biochemical Markers of Cardiac Remodeling

***TGF-β:*** In the LV of untreated SHR, mRNA expression of TGF-β_1_ was 2.68 [2.10/2.74] percent of GAP-DH. Treatments with CAP and CAP + NIF decreased TGF-β_1_ to 1.77 [1.64/2.11] and 1.19 [1.14/1.98], respectively, but this effect was significant only with CAP + NIF (*p* = 0.04). TGF-β_2_ and TGF-β_3_ were not significantly reduced with either treatment (Figure 3a–c). The changes in TGF-β_1_ and TGF-β_3_ were dependent on SBP at baseline.

However, both therapies significantly reduced the protein concentrations of TGF-β_1_ and TGF-β_2_ in the LV. TGF-β_1_ concentration decreased by 33% (CAP) and 21% (CAP + NIF) compared to CTRL (*p* = 0.03). For TGF-β_2_, the combination therapy proved to be more effective than monotherapy (*p* < 0.001), and reduced the protein concentration by 41% (CAP) and by 60% (CAP + NIF) compared to CTRL (*p* < 0.001; Figure 3d,e).

***MMP-2 and TIMP-2:*** The expression of MMP-2 and TIMP-2 mRNA in the LV of CTRL rats was 5.58% [4.43/8.98] and 24.9% [13.7/28.8], respectively. Both therapies resulted in a significant reduction of mRNA expressions of MMP-2 (CAP by 53%, CAP + NIF by 46% of CTRL; *p* = 0.001) and TIMP-2 (CAP by 57%, CAP by 59% of CTRL; *p* < 0.001). Both treatments also decreased the activity of MMP-2 by 41% (CAP) and 51% (CAP + NIF) of CTRL (*p* = 0.03; Figure 4). ANCOVA results showed that these changes did not depend on age, BW and SBP at baseline.

***Collagens:*** In the LV of CTRL rats, mRNA expression of Coll I and Coll III was 13.8% [9.7/16.7] and 24.8% [22.4/26.8], respectively. Both treatments significantly reduced Coll I to 4.5–4.8% (i.e., about 34% of CTRL, *p* < 0.001) and Coll III to 9.5–10.5% (i.e., about 40% of CTRL, *p* < 0.001; Figure 5). ANCOVA revealed that the effect of the treatments was not dependent on the values of the covariates age, BW and SBP at baseline.

#### 3.3.2. Histological Manifestation of Cardiac Fibrosis

The hearts of the untreated CTRL rats showed a marked interstitial and perivascular fibrosis (fibrosis degree 2.3 ± 0.11; Figure 6A). Both treatments significantly reduced the degree of perivascular and interstitial fibrosis. CAP therapy (fibrosis degree 1.8 ± 0.05; *p* = 0.001 vs. CTRL) proved to be slightly superior to CAP + NIF treatment (fibrosis degree 2.1 ± 0.07; *p* < 0.001 vs. CAP; Figure 6B,C). The histological degree of fibrosis did not depend on baseline age (about 60 weeks), but there was an interaction with BW and SBP at baseline.

## 4. Discussion

In the present study, we have shown that even a late start of antihypertensive therapy can achieve a significant reduction of SBP in old SHR with the long persistence of hypertension. At that age, cardiac hypertrophy, cardiac remodeling and fibrosis, as consequences of hypertension, have already developed; they could not be prevented but their progression was delayed and attenuated by both therapies. This corresponds to the functional effects of late-onset CAP and CAP + NIF treatments in old SHR as detected by echocardiography. Compared to untreated old SHR, both types of treatment significantly delayed the increase in LV wall thickness and preserved the systolic and diastolic pumping function for a significantly longer period of time [21].

### 4.1. Development of Hypertension in SHR in Early and Later Stages of Life

In SHR, hypertension develops in a very early period of life. Seven-week-old SHR showed 50% higher SBP values than normotensive Wistar-Kyoto rats (WKY) of the same age, and during the following three weeks, their SBP increased significantly to almost 170% of WKY [14]. During this early stage, SHR developed functional and morphological vascular changes [22,23] which are associated with vascular hypertrophy and increasing vascular resistance, thus leading to cardiac pressure overload [24]. The development of cardiac hypertrophy in SHR starts between four [25] and twelve weeks of age [26]. Our previous study on young SHR showed that mild cardiac hypertrophy and fibrosis were already present at 10 weeks of age [14]. As hypertension progresses, SHR develops increasing structural changes in the heart between the sixth and 24th months of life, which are associated with a further increase in cardiac hypertrophy [27]. In the context of hypertrophy and remodeling, numerous fetal genes are re-expressed, such as ANP [28,29].

The transition to heart failure in SHR is associated with the development of fibrosis resulting from the increased expression of genes encoding extracellular matrix (ECM) proteins and from the increased turnover of matrix proteins through increased MMP activity [20]. TGF-β as an important regulator of MMPs and TIMPs plays a crucial role in ECM remodeling [30,31]. The upregulation of TGF-β in the LV, particularly of TGF-β_1_, is associated with increased activity of MMP-2 and enhanced synthesis of collagens and other extracellular matrix proteins [4,8]. Of note, activation of the RAAS upregulates TGF-β_1_ synthesis and results in myocardial fibrosis. Correspondingly, RAAS antagonization attenuates these processes [8], which was also confirmed by the present results.

Significant collagen deposition in the heart of SHR has been found as early as at 8 weeks of age [32], and its amount further increased during adulthood [33]. The remodeling processes led to reduced contractile myocardial function, ventricular fibrosis and heart failure at an advanced age (>20 months) in untreated SHR [4,34]. This was also confirmed in an echocardiography study on old SHR. Both echocardiographic examination and heart catheterization showed marked LV dysfunction in untreated animals [21]. We assume that the loss in BW observed in the SHR CTRL group of the present study might be a consequence of this severe cardiac dysfunction. A comparison of the present results with those of the previous study in young SHR [14] reveals significant increases both in SBP and in biochemical and histological markers of ECM remodeling in the old SHR CTRL group (Table 1). In particular, it confirms a significant increase in markers of cardiac hypertrophy (HW/BW, ANP mRNA expression). Note that these markers did not depend on age, BW and SBP at the beginning of the experiment. The expression of TGF-β isoforms and TIMP-2 was also significantly higher in old than in young SHR, while MMP-2 expression was only slightly elevated. Coll I and III mRNA were equal in old and young SHR. However, considering the significantly higher histological degree of fibrosis in old SHR and the marked effects of treatment both in young and old SHR, this finding may suggest that collagen mRNA expression in SHR is elevated from a young age and does not adequately reflect the degree of collagen deposition in the heart. This corresponds with our observation that collagen mRNA in 82-week-old SHR is not dependent on age, BW, and SBP at baseline.

Data from young SHR are from [14]. Normally distributed parameters are given as mean ± SEM, and statistics were performed using a t-test. Parameters that were not normally distributed are given as median [25th/75th percentile], statistics were calculated using Mann-Whitney’s U test. HW/BW heart weight-to-body weight ratio; ANP atrial natriuretic peptide; TGF-β transforming growth factor-β; MMP-2 matrix metalloproteinase 2; TIMP-2 tissue inhibitor of metalloproteinases 2; Coll I, Coll III collagen types I and III.

### 4.2. Effects of Antihypertensive Therapies 

The degree of BP reduction is associated with the age at which treatment is implemented. The early start of antihypertensive therapy does not only reduce BP but has also cardioprotective effects, which are well-documented [2,7,13]. Early antihypertensive treatment can lead to normotension, whereas late therapy-onset results only in incomplete blood pressure regulation [35]. Moreover, early antihypertensive therapy can prevent or, at least, minimize secondary damage such as cardiac hypertrophy and remodeling, which again maintain and increase hypertension [2,3]. This was fully confirmed in our previous study on young SHR [14]. Harrap and co-workers [36] suspected a critical phase with an increased sensitivity to pharmacological influences between the 6th and 10th week of life in the development of hypertension in SHR. Intervention in the blood pressure regulation during this time may change the long-term course of disease development [36]. In young rats, RAAS antagonists are particularly effective, even if the therapy is stopped after a defined period of treatment [3,35,36]. In contrast, therapy onset at a higher age cannot completely reverse the development of hypertension. A study of Kost et al. [35] showed that SHR, which received captopril at the age of 24 weeks, had higher BP values than SHR treated with captopril at the age of four weeks. These results are in line with the findings of the present study: In old SHR, therapy with CAP or CAP + NIF over 22 weeks significantly reduced SBP by about 20%. However, these values were with 183 and 194 mmHg, respectively, about 25% higher than in 10-week-old SHR after only three weeks with CAP or CAP + NIF treatment [14]. Of note, in that study on young SHR, the relative SBP reduction (in % of baseline) was with 18% and 3%, respectively, lower than in the present old SHR study with the same doses of CAP and CAP + NIF. This clearly shows that the doses of antihypertensive drugs are also effective in old SHR indicating that their final SBP remains elevated mainly due to their higher baseline SBP. An interesting question is whether increased doses of antihypertensive drugs might induce a stronger SBP reduction.

There are only a few therapy studies in old SHR with late onset of antihypertensive treatment. Ito et al. [37], Susic et al. [38] and Mukherjee and Sen [39] examined old SHR at the age between 12 to 22 months during treatment with RAAS antagonists over different time intervals (between 4 and 12 weeks) and showed reduced signs for hypertrophy. Only Brooks et al. [40] treated SHR aged 12, 18 and 21 months with captopril 2 g/L in drinking water over a very long period of four to 12 months. They showed attenuated signs of heart failure and decreased ventricular hypertrophy even when heart failure had already started to develop.

In the present study, antihypertensive therapy with CAP or CAP + NIF over 22 weeks significantly reduced HW and HW/BW. Correspondingly, mRNA expression of the hypertrophy marker ANP in the LV decreased by 77 and 84%, respectively, compared to untreated CTRL (Figure 2). However, HW/BW was still about 12–14% above the values of 10-week-old SHR treated with CAP and CAP + NIF over three weeks. ANP mRNA of treated old SHR was even about tenfold of the values in corresponding young animals [14].

In addition, both types of treatment diminished ECM remodeling in the heart of old SHR. AT II plays an important role in the development of hypertension-induced cardiac fibrosis [8,41] by stimulating the production of IL-6 and TGF-β_1_ in cardiac fibroblasts [42]. A significant reduction of TGF-β_1_ expression by CAP was also shown by Brooks and co-workers [40] in 12 to 24-month-old SHR. As a result of the reduced TGF-β production, which was predominantly due to the effect of CAP, we observed a significant reduction of mRNA expressions of MMP-2, TIMP-2, Coll I and Coll III, as well as of MMP-2 activity by about 50–60% in the LV of old SHR (Figure 3, Figure 4 and Figure 5). Consequently, a significant decrease of histological signs of fibrosis was found in CAP and CAP + NIF animals (Figure 6). However, this effect was much lower than in young SHR: in our previous study on young SHR, fibrosis was attenuated by 60% with CAP + NIF treatment and completely abrogated by CAP treatment [14]. In contrast, in old SHR the fibrosis degree only decreased by 9% and 22%, respectively, despite a much longer period of treatment. As indicated by ANCOVA results, the high SBP at the beginning of treatment might have contributed to this reduced therapy effect. Consequently, functional deterioration due to fibrotic impediment could be delayed but not completely prevented.

An echocardiographic study in old SHR demonstrated that with antihypertensive treatment, systolic and diastolic pump function in treated SHR was maintained for a longer period of time. In these animals, a slight deterioration of cardiac function began around the 76th week of life, while untreated rats showed a progressive decline of systolic and diastolic function throughout the experimental period [21]. Once structural changes in LV have occurred, therapy with inhibitors of AT converting enzyme or calcium antagonists can no longer reverse them but delay and attenuate the deterioration of cardiac function [43]. Nevertheless, even in long-standing hypertension, which may be diagnosed late, immediate therapy can significantly lower SBP, delay the development of hypertrophy and significantly reduce fibrosis, as demonstrated in the present study. Thus, the onset of heart failure can be prevented or at least delayed.

### 4.3. Comparison of Monotherapy CAP with Combination Therapy CAP + NIF

Previous studies in rats have shown that RAAS antagonists have superior antihypertensive effects compared to other therapies, e.g., calcium antagonists [44]. RAAS antagonists play an important role in lowering HW, preventing cardiac hypertrophy and mitigating structural remodeling of the cardiovascular system and terminal organ damage in young SHR [2,7,13]. In our previous study on young SHR, a monotherapy with the calcium antagonist NIF proved to be effective, but was less effective than CAP [14].

In the present study on old SHR, monotherapy with CAP and the combination of CAP + NIF proved to be equivalent. Previous results of preliminary examinations in young SHR showed a particularly high antihypertensive and cardioprotective effect of therapies with RAAS antagonists [14]. Two therapies from that study were applied in the present study to old SHR (CAP and CAP + NIF). These therapeutic regimens induced a significant reduction in SBP and an attenuated and delayed development of cardiac hypertrophy. An echocardiographic examination in old SHR demonstrated that rats treated with CAP or CAP + NIF exhibited a stable LV function, while untreated SHR showed clear signs of LV insufficiency [21].

### 4.4. Limitations of the Study

The main limitation of this study is the low number of experimental animals. The group size was calculated for a medium effect size of SBP reduction (with Cohen’s f = 0.25). However, clinical symptoms of decompensated heart failure and also the BW reduction of CTRL rats showed high interindividual differences. These characteristics correspond to observations in human pathology. The number of animals to be examined should be kept as low as possible, since only as little SHR as necessary should be exposed to the suffering caused by the consequences of hypertension. The calculated sample size of 21 animals was considered representative.

A further limitation is that we did not measure tibia length. Treated SHR slightly increased their body weight over the experimental period, while untreated CTRL animals showed a weight loss of 11% relative to baseline. Thus, the ratio of heart weight to tibia length would be more adequate than HW/BW in reflecting cardiac hypertrophy.

With respect to the similar efficacy of CAP monotherapy and CAP + NIF combination therapy, further studies should include a direct comparison of monotherapies with CAP and with NIF in old SHR. Further research might also examine the question of whether a late-onset therapy with higher doses of antihypertensive drugs may result in the further reduction of SBP and better cardioprotective effects.

## 5. Conclusions

Two therapy regimes, a monotherapy with CAP and a combination therapy with CAP + NIF, proved to be equally effective at reducing SBP and delaying the development of cardiac secondary damage in old SHR. Long-term-treatment significantly reduced SBP as well as cardiac hypertrophy and fibrosis. However, the levels of young SHR with early-starting treatment were not achieved. These findings are of importance with regard to the late diagnosis of arterial hypertension in humans, which is often made in the elderly. Consequent antihypertensive therapy, even if it starts at an advanced age, can reduce cardiac stress and thus cardiac secondary diseases.

## Figures and Tables

**Figure 1 biomedicines-10-01964-f001:**
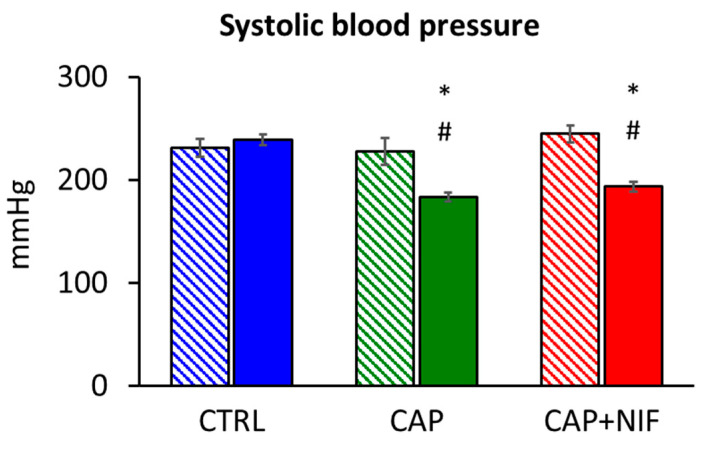
Systolic blood pressure of old SHR expressed as mean ± SEM at baseline (60 weeks of age; hatched columns) and at the final measurement after 22 weeks of experiment (82 weeks of age; filled columns). Significance marks: # significant vs. baseline measurement (*p* < 0.001); * significant vs. CTRL (*p* < 0.001).

**Figure 2 biomedicines-10-01964-f002:**
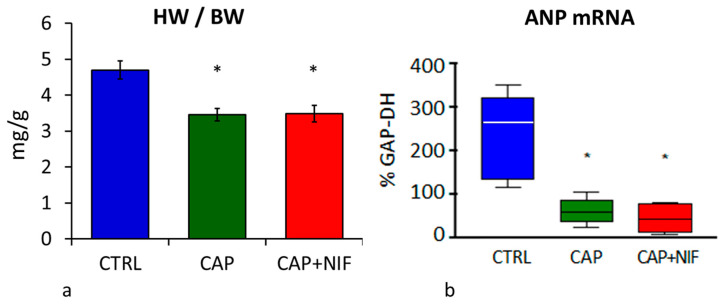
(**a**): Heart weight-to-body weight ratio (HW/BW; in mg/g) expressed as mean ± SEM; (**b**): mRNA expression of atrial natriuretic peptide (ANP, in % of GAP-DH) expressed as median (line in the box) with 25th/75th percentiles (boxes) and 10th/90th percentiles (whiskers). All values were obtained from 82-week-old SHR. Significance marks: * significant vs. CTRL (*p* ≤ 0.001).

**Figure 3 biomedicines-10-01964-f003:**
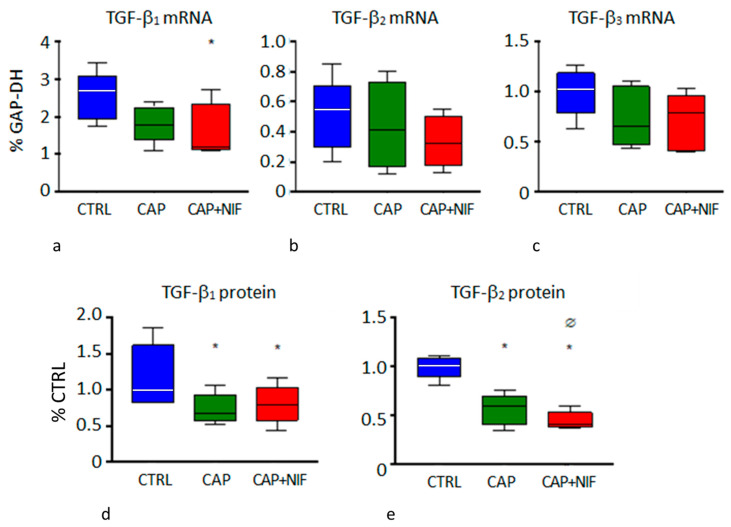
(**a**–**c**): mRNA expression of TGF-β isoforms (in % of GAP-DH) in the LV. (**d**,**e**): Protein concentrations of TGF-β_1_ and TGF-β_2_ (in % of CTRL) in the LV. Data are given as median (line in the box) with 25th/75th percentiles (boxes) and 10th/90th percentiles (whiskers). All values were obtained from 82-week-old SHR. Significance marks: * significant vs. CTRL; Ø significant vs. CAP.

**Figure 4 biomedicines-10-01964-f004:**
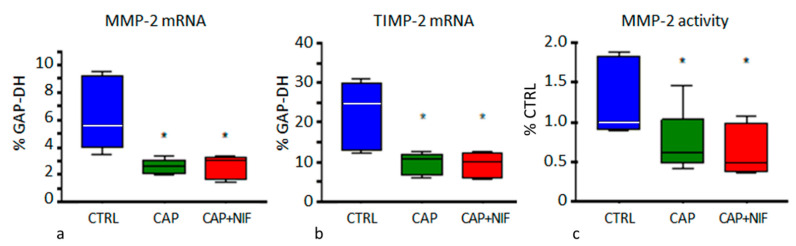
(**a**,**b**): MMP-2- and TIMP2 mRNA expression (in % of GAP-DH) in the LV. (**c**): MMP-2 activity (in % of CTRL) in the LV. Data are given as median (line in the box) with 25th/75th percentiles (boxes) and 10th/90th percentiles (whiskers). All values were obtained from 82-week-old SHR. Significance marks: * significant vs. CTRL.

**Figure 5 biomedicines-10-01964-f005:**
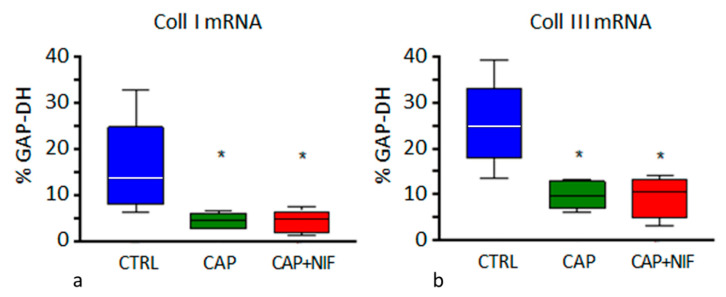
Coll I (**a**) and Coll III (**b**) mRNA expression (in % of GAP-DH) in the LV. Data are given as median (line in the box) with 25th/75th percentiles (boxes) and 10th/90th percentiles (whiskers). All values were obtained from 82-week-old SHR. Significance marks: * significant vs. CTRL (*p* < 0.001).

**Figure 6 biomedicines-10-01964-f006:**
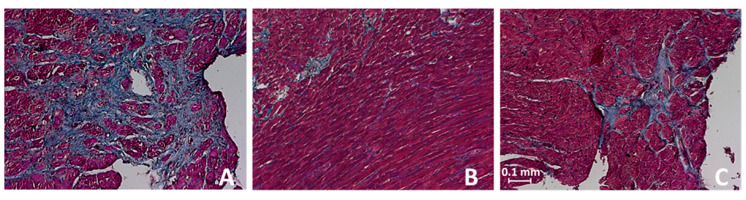
Hearts of 82-week-old CTRL and treated SHR in trichrome staining (10-fold magnification; the scale bar in part C represents 0.1 mm). (**A**): CTRL, perivascular and marked interstitial fibrosis, fibrosis degree of the group 2.3 ± 0.11; (**B**): CAP, perivascular and slight interstitial fibrosis, fibrosis degree of the group 1.8 ± 0.05; (**C**): CAP + NIF, perivascular and interstitial fibrosis, fibrosis degree of the group 2.1 ± 0.07 (data are given as means ± SEM).

**Table 1 biomedicines-10-01964-t001:** Comparison of systolic blood pressure (SBP) and parameters of cardiac hypertrophy and ECM remodeling in young (10-week-old) and old (82-week-old) SHR.

Parameter	Young SHR (10 Weeks)	Old SHR (82 Weeks)	*p* Value
final SBP	202 ± 6	239 ± 5	0.002
HW/BW	3.50 ± 0.07	4.70 ± 0.25	0.003
ANP mRNA	11.0 [7.5/20.3]	264 [154/293]	0.008
TGF-β1 mRNA	1.37 [1.34/1.54]	2.68 [2.10/2.74]	0.004
TGF-β2 mRNA	0.16 [0.14/0.18]	0.56 [0.55/0.72]	0.004
TGF-β3 mRNA	0.64 [0.54/0.67]	1.02 [0.93/1.12]	0.02
MMP-2 mRNA	3.92 [3.57/4.69]	5.58 [4.43/8.98]	0.222
TIMP-2 mRNA	8.01 [6.95/10.2]	24.9 [13.7/28.8]	0.008
Coll I mRNA	16.3 [9.6/16.6]	13.8 [9.7/16.7]	1.0
Coll III mRNA	20.9 [20.1/22.4]	24.8 [22.4/26.8]	0.548
Degree of fibrosis	1.47 ± 0.12	2.30 ± 0.11	<0.001

## Data Availability

Not applicable.

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
