# Peer review of "How Effective Is a Late-Onset Antihypertensive Treatment? Studies with Captopril as Monotherapy and in Combination with Nifedipine in Old Spontaneously Hypertensive Rats"

_biomedicines, 2022, doi:10.3390/biomedicines10081964_

Round 1
Reviewer 1 Report
To evaluate the effectiveness of a late-onset antihypertensive treatment, this author conducted an experimental study of 21 spontaneously hypertensive rat. From this study, this author found that despite the late start of therapy, cardiac hypertrophy and fibrosis were significantly attenuated. As a reviewer, I think this manuscript deal with interesting topic and results were informative. However, only few issues raised my concern. I commented those as following.
Abstract
1. Since all rat used in this study is male rat, this information should be shown in abstract section.
2. In result section of abstract, this author used the sentence of “significantly”. However, there is no data of p value.
3. Because this is a scientific journal, specific values should be shown. For example, if this author wants to compare the decreasing level of blood pressure between taking anti-hypertension medication group and untreated group, blood pressure change between baseline and endpoint should be compared between those groups. And p vales also should be shown.
4. In abstract, only crude data were shown. However, age and weight adjusted value should be used to evaluate the present variables.
5. Only Information from abstract, this information never brought present conclusion as I commented above. Abstract should be described in logically.
Introduction section
6. This author should describe their hypothesis of present study. Such a description could made readers to read present manuscript much easier.
Statistical analysis
7. Weight at baseline might influence on the results measure at endpoint. And age in days also might influence on present results. Then baseline weight and age in days adjusted value of present main results should be shown.
Discussion
8. To evaluate the effectiveness of a late-onset antihypertensive treatment is the main purpose of present study. However, in present study, there is no data that shows the effectiveness of an early-onset antihypertensive treatment. This should be a limitation of present study.
Author Response
We thank this reviewer for his/her valuable comments that helped to improve the quality of the manuscript. Please find our detailed responses to the comments in the attached file.

Reviewer 2 Report
In this study, the authors sought to investigate whether late-onset antihypertensive treatment is effective in lowering blood pressure and in reducing left ventricular hypertrophy and fibrosis.
The study is well designed and results are straightforward and clearly presented. Overall, the manuscript is well written. I have following comments:
In hypertensive patients, treatment is initiated upon diagnosis independent of the age and duration of the disease. A significant proportion of untreated hypertensive patients might be related to their low compliance or unawareness of the disease, and not due to the assumption by doctors of low efficacy of late-onset treatment. In this regard, the rationale for the study is not clear.
Given that the dose of captopril was the same in the monotherapy and combination therapy groups, no difference in the treatment effects between these groups suggests that nifedipine was probably inefficient. Inclusion of another group of monotherapy with nifedipine would be appropriate.
The authors state that the SBP values and heart hypertrophy in old SHR following therapy were still higher than those in young SHR from a previous study. However, the SBP values and degree of heart hypertrophy in control old SHR were also significantly higher than those in control young SHR. Adjusting therapy by increasing the doses of the drugs could induce further reduction in SBP and attenuation of heart hypertrophy.
Lines 118-119. What was the reason for the different body weight gain between the groups? In this case, the more appropriate measure of heart weight in old animals would be ratio of heart weight to tibia length.
Lines 258-259. Treatment was initiated in SHR at age of 60 weeks. According to the literature, at this age, heart hypertrophy has already developed. Therefore, it is not correct to state that late-onset treatment prevented heart hypertrophy in these rats.
Line 291. Please correct the “ANF” to “ANP”
Line 299. In the table title, “they are cited” is not clear
Lines 385-386. Sample size calculation would be a more appropriate way to keep the number of animals low.
Author Response
We thank this reviewer for his/her valuable comments that helped to improve the quality of the manuscript. Please find our detailed responses to the specific comments in the attached file.

Round 2
Reviewer 1 Report
Thanks to this author’s great effort, present manuscript became well improved. However, as I commented before, this author still only used crude model in statistical analysis.
As this author described in response, the average age of the animal at baseline is 60.5 ±0.25week. Therefore, there is 0.25 weeks of 1 standard deviation. Since age, weight and blood pressure at baseline could influence on present results, those factors should be adjusted in present study by using analysis of covariance. Even this author used relative heart weight (body weight related heart weight) this never supports that adjustments for baseline clinical characteristics are unnecessary. Relative heart weight indicates the weight related heart weight at the time that was measured. Then, present analysis never adjusted for baseline confounding factors. Results from such an analysis, this author could not conclude because baseline characteristics but not medication itself could lead present results.
Author Response
In the second round, reviewer #1 insisted, that analysis of covariance should be done. He/she argued, that age, weight and blood pressure at baseline might have had an influence on the results. We have now performed analysis of covariance and have included the results in the individual paragraphs of the manuscript (in blue). Except for the changes in TGFß1- and TGFß3-mRNA and histological degree of fibrosis, all other changes were not dependent on age, weight and blood pressure at baseline. Thus, they were the result of the therapeutic interventions. We appreciate the persistent, constructive and valuable critique of reviewer#1.
Round 3
Reviewer 1 Report
Thanks to this author’s great effort, present manuscript holds enough value to be published.
I have no more recommendation on this manuscript.